# Effect of Calanus Oil Supplementation and 16 Week Exercise Program on Selected Fitness Parameters in Older Women

**DOI:** 10.3390/nu12020481

**Published:** 2020-02-14

**Authors:** Klára Daďová, Miroslav Petr, Michal Šteffl, Lenka Sontáková, Martin Chlumský, Miloš Matouš, Vladimír Štich, Marek Štěpán, Michaela Šiklová

**Affiliations:** 1Faculty of Physical Education and Sport, Charles University, 162 52 Prague, Czech Republic; petr@ftvs.cuni.cz (M.P.); steffl@ftvs.cuni.cz (M.Š.); Sontakova.L@seznam.cz (L.S.); mcedok@gmail.com (M.C.); 2Third Faculty of Medicine, Charles University, 110 00 Prague, Czech Republic; milosmatous@seznam.cz (M.M.); vladimir.stich@lf3.cuni.cz (V.Š.); marek.stepan89@gmail.com (M.Š.); michaela.siklova@lf3.cuni.cz (M.Š.); 32nd Internal Medicine Department, University Hospital Královské Vinohrady, 100 34 Prague, Czech Republic

**Keywords:** older age, circuit training, nordic walking, calanus oil

## Abstract

We investigated changes in functional fitness after an exercise program in combination with Calanus oil supplementation, a novel source of bioactive lipids rich in wax esters with omega-3 polyunsaturated fatty acid (n-3 PUFA). Fifty-five healthy sedentary women aged 65–80 (mean age 70.9 ± 3.9 years, BMI 27.24 ± 3.9 kg m^−2^, VO_2_peak 19.46 ± 3.7 ml kg^−1^ min^−1^) were enrolled in the study. The participants were divided into two groups: exercise training plus Calanus Oil supplementation (*n* = 28) or exercise plus placebo (sunflower oil) supplementation (*n* = 27). The exercise intervention program was completed by 53 participants and contained functional circuit training (twice a week, 45 min plus 15 min of stretching and balance training) and Nordic walking (once a week, 60 min) for 16 weeks. Senior fitness test, exercise stress test on bicycle ergometer, hand-grip, and body composition were evaluated before and after the program. Our results show that functional fitness and body composition improved following the interventional exercise program, but for most of the parameters there was no synergic effect of supplementing n-3 PUFA-rich Calanus oil. In comparison to the placebo group, the group with Calanus supplementation experienced significantly higher improvement of functional strength of lower body which was evaluated by the chair stand test. Supplementation with Calanus may have a synergic effect with exercise on functional strength of the lower body in the elderly.

## 1. Introduction

Aging is associated with loss of muscle mass and strength [1]. A progressive loss of muscle mass tends to accelerate after the age of 60–70 years [2,3]. Similarly, strength loss is faster after 70 years of age, where the decrease reaches 25–40% per decade in comparison to younger people, where it is usually less than 15% per decade [4].

Effects of regular exercise or long-term interventional exercise programs on age-related complications have been widely studied; positive adaptations include increased muscle mass and strength [5], decreased adiposity, and increased cardiorespiratory fitness [6]. On the other hand, the adverse effects of regular exercise in the elderly include muscle soreness or swollen joints in sedentary people unaccustomed to exercise [7].

However, nutrition is another crucial factor which affects muscle mass and function, and it seems to be critical for reaching a desirable anabolic response following resistance exercise in elders [8,9].

Besides the essential role of proteins in muscle protein synthesis or prevention of the loss of muscle mass [10], some studies suggest beneficial effects of several dietary compounds such as omega-3 fatty acids (n-3 PUFA) [11,12].

In a study by Smith et al. [11], the authors evaluated the efficacy of fish oil-derived n-3 PUFA only (without any exercise) on age-associated loss of muscle mass and function after six months supplementation in sixty healthy 60–85-year-old men and women. They demonstrated increased skeletal muscle mass, hand-grip strength, and 1-RM muscle strength in the n-3 PUFA supplemented group. Another group of authors showed that administration of 3.9 g/day for 16 weeks in older men and women reduced mitochondrial oxidant emissions, increased post-absorptive muscle protein synthesis, and enhanced anabolic responses to exercise [12].

Calanus Oil is a novel type of bioactive n-3 PUFA lipid, containing substantial levels of wax esters (up to 80%–90% of total lipids). It has been shown that Calanus is a highly bio-available source of eicosapentaenoic acid (EPA) and docosahexaenoic acid (DHA), fatty acids for human consumption [13] with a beneficial effect on inflammation and obesity related metabolic disturbances as shown in mice [14,15].

Furthermore, the combination of n-3 PUFA supplementation and exercise showed greater efficacy over exercise alone in several studies with aging individuals [16,17]. In the study of Da Boit et al., authors found that maximal isometric torque after exercise and muscle quality were mainly increased in the n-3 PUFA group compared with the placebo group in women, but not in men [16]. The exercise and omega-3 effect on muscle but also on other organs might be driven through the mechanism of reduced inflammation and improvements in the immune system [17,18].

Based on the surveyed literature, dietary n-3 PUFA in combination with exercise seems to be a promising strategy to sustain muscle mass and combat sarcopenia in the elderly [17,18,19]. Thus, our study aims to evaluate the effects of 16 weeks of exercise training and n-3 PUFA containing Calanus oil supplementation on muscle function and health related parameters in a group of older women in a randomized controlled trial.

## 2. Materials and Methods

### 2.1. Participants

Fifty-five healthy non-obese female participants aged 65–80 (70.9 ± 3.9 years, VO_2_peak 19.46 ± 3.7 ml kg^−1^ min^−1^) were enrolled in this study. Participants were recruited in the region of Prague (Czech Republic), at different educational events for older adults. Exclusion criteria involved: regular exercise more than once a week before the study, medical history of cancer, diabetes, liver or kidney disorders, long term medication with antirheumatic agents, steroids, and/or n-3 PUFA supplementation. Participants were randomly divided into one of the following groups: Calanus supplementation plus exercise training (Calanus, *N* = 28) and placebo supplementation plus exercise training (placebo, *N* = 27). The study was conducted as double-blinded trial. Two participants withdrew from the study during the intervention, because of newly diagnosed health problems that were not caused by the exercise intervention itself. Thus 27 participants in the Calanus group and 26 participants in the placebo group were finally analyzed. The groups did not differ in standard parameters such as body mass index (BMI), age, or cardiorespiratory fitness (measured as peak oxygen consumption). In the study, participants consumed daily either five capsules of Calanus oil (Calanus) or sunflower oil (placebo). The dose of n-3 PUFA in the Calanus group was approximately 125 mg eicosapentaenoic acid (EPA) and 105 mg of docosahexaenoic acid (DHA) per day. Participants kept three-day dietary records before and after the study to monitor their diet. Furthermore, the questionnaires for monitoring n-3 PUFA consumption were filled before and after the intervention. The n-3 PUFA consumption and dietary records of participants were not changed during the study. The Ethical Committee of the University Hospital Královské Vinohrady in Prague approved the study. All participants signed informed consent before the start of the study for voluntary participation in this study; all data were anonymized. Participants were financially rewarded for the study participation. The study was part of EXODYA (effect of exercise training and Omega-3 fatty acids on metabolic health and dysfunction of adipose tissue in elderly) research project (nr. AZV 16-29182A). Clinical Registration No: NCT03386461. The data were collected at baseline and after the exercise intervention program (pre-post test).

### 2.2. Anthropometry and Body Composition

Body height was measured using a SECA 213 portable stadiometer, body mass was measured using a SECA 876 digital flat floor scale, and body composition (fat and muscle mass) and related parameters (intracellular water, ICV; extracellular water, ECW; visceral fat, VFA) were measured via bioelectrical impedance (InBody 720, Biospace Co., Ltd., Seoul, Korea).

### 2.3. Assessment of Exercise Capacity

Each subject performed an exercise stress test (EST) on a bicycle ergometer (Ergoselect 200, Ergoline) consisting of one grade of submaximal exercise at an intensity of 0.5 Watts per kilogram of body weight (W kg^–1^), lasting 4 min and a continuously increased exercise stress test (“ramping protocol”). The load increment was linear, beginning at a load of 20 W with an individually chosen slope of 15–30 W per minute in order to achieve the expected peak load within 4 to 8 min. Participants were required to maintain a revolution rate of between 70 and 90 per minute throughout the test while they were encouraged verbally to reach the point of exhaustion. An individual peak effort was identified, when the subject’s respiratory exchange ratio (RER) value reached more than 1.06, and the subject was unable to maintain a constant pedaling rate despite the effort. None of the participants interrupted EST before the peak effort level as a result of cardiorespiratory difficulties or other medical complications. Before and throughout the test, values of HR, blood pressure (BP), and electrocardiography (ECG, Schiller) were recorded. Oxygen consumption (using a breath-by-breath gas exchange analyzer Power Cube-Ergo (Ganshorn Medizine Electronic GmbH, Niederlauer, Germany) was measured in the last minute of submaximal exercise and during the entire ramp protocol. Peak values of oxygen consumption and the highest power output attained were used to describe exercise capacity.

### 2.4. Assessment of Functional Fitness

The senior fitness test (SFT) is a test battery assessing major functional fitness components of adults aged 60+ [20]. Functional fitness is defined as the ability to perform everyday tasks (the activity of daily living, instrumental activities of daily living) with adequate energy, without undue fatigue and with sufficient reserve [21]. SFT is a standardized test with good content and criterion validity and high test-retest reliability [21]. Each participant completed four motor tests from the SFT battery in the following testing order: 30 s chair stand test, arm curl test, sit and reach test, and back scratch test. Tests were administered before and after the 16-week duration exercise intervention (i.e., pre-post test).

The 30 s chair stand test assesses functional lower body strength. The tested person performs as many full stands from a fully seated position on a chair (height of 43 cm) for 30 s, with arms folded across the chest. Only one test trial is allowed.

The arm curl test assesses functional upper body strength. The seated person performs a bicep curl in full range of motion from a fully extended dominant hand with a hand weight (women 2.27 kg) for 30 s. Only one test trial is allowed.

The sit and reach test measures the flexibility of lower back and hamstring muscles. From a seated position at the front of a chair (height of 43 cm), the tested person bends forward while reaching forward to a fully extended preferred leg with the heel on the floor and the foot flexed at a right angle. The other leg is bent slightly to the side, foot flat on the floor. The maximum reach must be held for 2 s. The distance in cm between the tip of the middle fingers and the shoe is measured. Scoring is as follows: a minus (−) score if the reach is short, a plus (+) score if the fingers reach behind the toes, a zero (0) if the middle fingers touch the toes. After two practice trials on the preferred leg, two test trials are administered, with the best value used.

The back scratch test measures the flexibility of the upper body and shoulders, with one hand reaching over the shoulder and one up the middle of the back with measurement of the number of cm between the extended middle fingers. The maximum reach must be held for 2 s. The distance in cm between the tip of the middle fingers is measured. Scoring: a minus score (−) if the reach is short, a plus (+) score if the middle fingers overlap, a zero score (0) when the middle fingers barely touch. After two practice trials on a preferred arm, two test trials are administered. The best value from the two test trials is used.

### 2.5. Assessment of Strength of Hand Muscles

The strength of hand muscles (hand-grip) was measured by digital dynamometer TKK 5401 (Takei, Japan) according to standard methodology. The best value from the three trials was used.

### 2.6. Exercise Intervention

The duration of intervention was 16 weeks between February and May in 2017 and 2018. The exercise program involved a combination of aerobic and resistance training and included three group exercise lessons led by a certified instructor each week. For the first two weeks, lessons focused mainly on proper posture, coordination, breathing, stability, flexibility, and acquiring exercise skills and habits. This acclimatization period was applied in order to learn the correct techniques of exercise. After this period, lessons started to include strength training (circuit training) in a gym twice a week and aerobic training (i.e., Nordic walking) once a week. Furthermore, participants were encouraged to increase their habitual physical activity.

#### 2.6.1. Strength Training Lessons

The duration of each exercise lesson in the gym was approximately 60 min, including 15 min warm up followed by 45 min circuit training (2–3 rounds of 8–10 exercises with a duration of 45–60 s each, and rest between them of 20 s). Exercises were based on functional strength training adapted for the elderly, using their own weight and equipment like dumbbells, Therabands, Bosu balls, stability balls, total resistance exercise (TRXs), steppers, etc. Participants were encouraged to exercise at a moderate intensity of RPE 13–14 according to the original 15-grade Borg rating of perceived exertion scale [22]. The whole exercise lesson was completed with calming and stretching exercises (about 15 min).

#### 2.6.2. Aerobic Training Lessons

Aerobic training lessons were based on Nordic walking trips in a close neighborhood surrounding the training facility. Each Nordic walking lesson included a warming up period for 10 min at a slower pace, a little stretching, the main part consisted of walking at a faster pace for about 40 min, and the 10 min cooling down period was walking at a slower pace. During the main period of aerobic training, participants were encouraged to walk at moderate to high intensity (i.e., 60%–85% VO_2_peak), individually calculated from the initial stress test. HR was checked both by a telemetric device (Sporttester InSPORTline Diverz, Seven Sport, Czech Republic) and by the palpation method. For most of the time, the walking speed was at the lower level of the above-mentioned range. During all the walking trips, there were several short sections of higher intensity (i.e., interval training).

#### 2.6.3. Exercise Adherence

The mean adherence expressed for exercise attendance (% of attended lessons from all 46 lessons planned) for the whole group was 96.3% (range 74%–100%).

### 2.7. Statistical Analysis

Data are presented as median and interquartile range (IQR). As most of the data did not meet the standard criteria of normality, non-parametric tests were used for the evaluation (Mann–Whitney U test, Wilcoxon signed rank test, Spearman rank correlation). Results were considered statistically significant if *p* < 0.05. For the effect size, we calculated a correlation coefficient r using the Z value from the Mann-Whitney U test as *r* = Z/√N, where N is the total number of the participants. Standard interpretation for r is the same according to Cohen’s classification of effect sizes where 0.1 = small effect, 0.3 = moderate effect and ≥0.5 = large effect. All the statistics were carried out in IBM SPSS Statistics 24.

## 3. Results

Table 1 shows the general characteristics of participants in both groups showing that there were no significant differences between the Calanus group and the placebo group before the exercise intervention program.

The intervention-induced changes of functional physical performance within and between both groups are presented in Table 2. Functional strength of lower body (chair stand), as well as of upper body (Arm Curl) improved in both groups significantly (*p* < 0.05) after the program. Lower back and hamstring flexibility (sit and reach) did not change after the program in either group. Upper body and shoulder flexibility (back scratch) was significantly improved in the placebo group only. The strength of hands (hand grip) improved significantly (*p* < 0.05) in the placebo group only. As for exercise capacity (peak oxygen consumption and peak workload), both groups improved significantly (*p* < 0.05). When comparing pre-post changes between the Calanus and placebo groups (i.e., the effect of supplementation), the only statistically significant difference was in the chair stand (*p* = 0.02).

There was a significant decrease in body weight and, consequently in BMI in the placebo group after the intervention program (Table 3), while in the Calanus group, there was a statistically significant decrease (*p* = 0.02) in visceral fat area (VFA). The placebo group also showed a decrease in this variable but without statistical significance. The exercise-induced changes in body composition did not differ between groups.

Furthermore, we also focused on the relationships between the above-mentioned variables, especially pre-post changes. There was a statistically significant (*p* < 0.05) correlation between the following variables: Chair stand pre-post change and arm curl pre-post change were positively related (*r* = 0.56). Percentile pre-post change in VO_2_peak was negatively related to the VO_2_peak at baseline (*r* = −0.5) and positively to skeletal muscle mass (SMM) at baseline (*r* = 0.38).

## 4. Discussion

The biological process of aging cannot be stopped by any known means, but there is a body of substantial evidence that regular physical activity can promote life expectancy and prevent or postpone common modern lifestyle diseases and thus increase the quality of life. As previously mentioned, the effects of regular exercise or long-term interventional exercise programs on the onset of age-related complications have been widely studied and have shown increases in muscle strength and cardiorespiratory fitness [5,6]. For targeting inflammation as a significant contributor to sarcopenia, which also may compromise the anabolic effect of exercise on skeletal muscle [23], the supplementation of n-3 PUFA is proposed as an alternative and safe treatment to anti-inflammatory drugs [18]. Anti-inflammatory drugs seem to inhibit muscle growth [24]. On the other hand, the safety of n-3 PUFA supplementation in healthy individuals has been acknowledged by relevant authorities, such as EFSA (supplemental intakes of EPA and DHA combined at doses of up to 5g/day do not raise concerns for adults) [25]. Because a combination of exercise and n-3 PUFA supplementation has shown some beneficial effects on health and fitness [16,18,19], we focused on a novel class of n-PUFA supplementation–Calanus Oil produced from the marine copepod *Calanus finmarchicus* [26]. Calanus oil contains a combination of fatty acids, fatty alcohols, and wax esters, and is possibly the best alternative to fish oil whose production cannot keep pace with demand from the growing market [27]. Although, higher doses of Calanus oil are likely to be needed to increase the omega-3 index (defined as the sum of EPA and DHA in erythrocyte membranes expressed as a percentage of total fatty acids) as a reflection of an adequate resource of n-3 PUFA, its bio-availability seems to be sufficient [13].

The beneficial effects of n-3 PUFA on physical fitness or resistance training were shown in several studies. Da Boit et al. [16] documented that long-chain n-3 PUFA (2.1g EPA/day + 0.6 g DHA/day) supplementation augmented increases in muscle function after lower-limb resistance exercise training with little concurrent effect on muscle mass. A 24-week healthy diet and exercise program published by Edholm et al. [19] showed that a diet enriched by n-3 PUFA and balanced in n-6 PUFA content could optimize the effect of resistance training on dynamic and explosive strength evaluated by squat jump, sit-to-stand, and single-leg-stance tests.

Our analysis of functional fitness and body composition changes following an interventional exercise program showed a beneficial effect of exercise, however, the synergic effect of supplementing Calanus on most of the parameters was not observed. The variable which was improved by Calanus oil was the functional strength of lower body (30 s chair stand test). Similarly, Rodacki et al. [28] showed that a strength training program for 90 days combined with fish oil supplementation increased muscle strength and functional capacity in older women. Supplemented groups demonstrated a more significant improvement in chair stand performance, which was the only functional muscle test included in their study to show a positive result from supplementation. It should be noted that the n-3 PUFA dosage in this study was higher (fish oil 2 g/day corresponding to 360 mg EPA and 240 mg DHA) when compared to our study. Indeed, the dose of 230 mg EPA and DHA per day in our study was at the lower level of the recommended effective dietary intake [29]. However, consumption of five capsules was chosen as the maximal dose for good adherence of the subject to the study. Despite the lower dose of EPA and DHA we observed some effect on functional lower body strength. Thus, it might be hypothesized that a longer duration or higher dose of the Calanus supplement could also achieve effects on other parameters, and this should be included in future studies.

Additionally, our study is somewhat more diverse in terms of physical activity compared to the studies mentioned above because it involved not only strength training (twice a week) but also aerobic exercise (Nordic walking once a week). We registered a significant decrease in body weight and BMI in the placebo group, however this did not occur in the Calanus group. Regardless, we can speculate that improved metabolic health occurred because there was a statistically significant decrease in visceral fat registered by bioelectrical impedance in the Calanus group. Meanwhile, the placebo group did not reach statistical significance in this variable. This finding is potentially crucial for clinical practice, although we are aware of the shortcomings of the bioelectrical impedance method that is not as accurate as for example dual-energy x-ray absorptiometry (DXA) and rather small effect size. In accordance with previous studies, the Calanus derived wax esters were shown to improve diet-induced obesity and metabolic parameters in mice [15]. Thus, Calanus oil supplementation together with physical exercise may be suggested as a preventive strategy in obesity treatment. Indeed, the objective of the EXODYA project, which our investigation is a part of, is to study metabolic health using sophisticated biochemical and metabolomic methods. In recently published article we have shown positive effects of this intervention on metabolic parameters [30]. 

Concerning the relationships between pre-post differences of some variables, we obtained some interesting findings. Improvement of functional strength in the upper body was positively related to the improvement of functional strength in the lower body showing that the program was somewhat complex, involving all muscle groups. Percentage improvement of the VO_2_peak was negatively related to baseline values, pointing at an increased probability for better physical fitness improvement when the participant is less fit in the pre-study period. Our results also clearly show that a better chance to improve aerobic capacity (measured as VO_2_peak) occurred in women who had higher SMM.

Improvement in functional capacity after the exercise program regardless of supplementation with Calanus Oil or not shows the importance of regular physical activity for fitness and possibly quality of life [31]. An additional or synergic influence of Calanus supplementation on exercise, even in a low dose, may have a positive impact on visceral fat and lower body strength which is most relevant for fall prevention and maintaining activities of daily living in older age. Demonstrably higher doses of Calanus may cause more convincing findings of previous studies. For this reason, preventive and treatment strategies to delay sarcopenia, including appropriate exercise and n-3 PUFA/Calanus supplementation, should be further explored.

## Figures and Tables

**Table 1 nutrients-12-00481-t001:** Descriptive statistics (baseline values).

	Calanus	Placebo	
	*n* = 27	*n* = 26	*p*-Value
Age (years)	71.0 (4.0)	71.0 (5.3)	0.879
Height (cm)	161.0 (9.0)	161.0 (9.5)	0.943
Weight (kg)	69.9 (9.3)	68.5 (19.3)	0.575
BMI (kg.m^−2^)	26.8 (4.9)	26.5 (5.9)	0.735

Note: data are presented as median (IQR); Mann-Whitney U test was used to test for differences; BMI—body mass index.

**Table 2 nutrients-12-00481-t002:** Comparison of physical performance within groups and between groups.

	Values ^a^				Absolute Changes ^b^		
	Calanus		Placebo		Calanus	Placebo	
	(*n* = 27)		(*n* = 26)		(*n* = 27)	(*n* = 26)	*r* ^†^
Parameters	Pre	Post	Pre	Post	Post-Pre	Post-Pre	
Chair Stand (repetitions)	16.0 (6.0)	20.0 (5.0) *	16.0 (5.0)	18.5 (5.0) *	4.0 (5.0)	3.0 (3.0) *	0.316
Arm Curl (repetitions)	20.0 (8.0)	26.0 (6.0) *	19.0 (6.3)	21.5 (6.3) *	6.0 (5.0)	3.0 (6.3)	0.255
Sit and Reach (cm)	12.0 (14.0)	13.0 (10.0)	11.5 (11.0)	12.0 (8.9)	2.0 (8.0)	1.0 (6.1)	0.105
Back Scratch (cm)	−2.0 (11.0)	−1.5 (13.0)	1.8 (10.1)	4.3 (9.3) *	1.0 (3.0)	1.5 (3.4)	0.176
Hand Grip (kg)	26.2 (4.7)	25.2 (4.3)	26.4 (5.5)	27.1 (6.2) *	1.0 (2.0)	0.9 (2.3)	0.054
Peak power output (W. kg^−1^)	1.6 (0.5)	1.7 (0.4) *	1.6 (0.5)	1.7 (0.3) *	0.1 (0.2)	0.2 (0.2)	−0.218
VO_2_peak (ml.kg^−1^.min^−1^)	18.8 (4.7)	21.2 (7.7) *	18.4 (5.3)	22.6 (5.2) *	2.4 (5.0)	3.4 (5.0)	−0.072

Note: data are presented as median (IQR), * *p* < 0.05, ^a^ Wilcoxon Signed Rank test, ^b^ Mann–Whitney U test; *r* = effect size; **^†^** minus prefers placebo; VO_2_peak—oxygen uptake during peak exercise.

**Table 3 nutrients-12-00481-t003:** Comparison of body composition within groups and between groups.

	Values ^a^				Absolute Changes ^b^		
	Calanus		Placebo		Calanus	Placebo	
	(*n* = 27)		(*n* = 26)		(*n* = 27)	(*n* = 26)	*r* ^†^
Parameters	Pre	Post	Pre	Post	Post-Pre	Post-Pre	
Weight (kg)	69.9 (9.3)	68.7 (8.1)	68.5 (19.3)	67.4 (18.7) *	−0.1 (1.9)	−1.1 (1.5)	−0.208
ICW (l)	19.9 (2.5)	20.0 (3.2)	20.1 (4.1)	20.0 (4.2)	0.0 (1.1)	0.1 (2.9)	0.038
ECW (l)	12.7 (1.7)	12.8 (1.9)	12.6 (2.4)	13.0 (2.5)	0.0 (0.5)	−0.1 (2.0)	0.062
Fat (kg)	26.5 (10.8)	24.5 (9.9)	23.3 (8.7)	22.0 (8.3)	−1.1 (3.1)	−0.7 (9.6)	0.007
Fat (%)	37.2 (7.7)	35.2 (7.8)	34.5 (10.0)	33.9 (8.4)	−0.8 (3.3)	−1.0 (8.0)	0.040
SMM (kg)	23.9 (3.2)	24.1 (4.1)	24.1 (5.3)	24.1 (5.5)	0.1 (1.3)	0.1 (3.4)	0.057
VFA (cm^2^)	114.8 (46.0)	98.2 (33.8) *	101.0 (43.0)	92.3 (45.5)	−13.6 (29.3)	−12.7 (27.4)	0.109
BMI (kg.m^−2^)	26.8 (4.9)	26.8 (4.4)	26.5 (5.9)	26.0 (5.7) *	0.0 (0.7)	−0.4 (0.7)	−0.197

Note: data are presented as median (IQR), * *p* < 0.05, ^a^ Wilcoxon Signed Rank test, ^b^ Mann–Whitney U test; *r* = effect size; **^†^** minus prefers placebo; ICW—intracellular water; ECW—extracellular water; SMM—skeletal muscle mass; VFA - visceral fat area; BMI—body mass index.

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
