# Peer review of "Effect of Calanus Oil Supplementation and 16 Week Exercise Program on Selected Fitness Parameters in Older Women"

_nutrients, 2020, doi:10.3390/nu12020481_

Round 1

Reviewer 1 Report

In the material and methods, the characteristics of the food consumed by the participants, amount, type, etc, must be expressed to avoid possible bias in the results related to the type of feeding of participants

Author Response

Dear reviewer,

thank you for your revision. We improved the method section.

We put our anwers in italics after your points.

In the material and methods, the characteristics of the food consumed by the participants, amount, type, etc, must be expressed to avoid possible bias in the results related to the type of feeding of participants

The statement „Subjects kept three-day dietary records before and after the study to monitor their diet. Furthermore, the questionnaires for monitoring n-3 PUFA consumption were filled before and after the intervention. The n-3 PUFA consumption and dietary records of subjects were not changed during the study.“ was added to the manuscript.

Reviewer 2 Report

Nutrients: Calanus oil Review

l.44: Please check your reference #10 to determine what type of lifting was done by the subjects to calculate 1-RM (e.g.; bench press, squat, or what) ? Also I would like to know the characteristics of the subjects, as 1-RM, in my best judgement, is only accurate and valid when testing advanced weight trainers. If the subjects were not in that category, I would skip the reference to that test or choose a better reference. l.51: Please indicate that the data collected in Ref.’s 13 and 14 was collected using a mouse model. l.140 My understanding is that RPE is on a scale of 1-10, so 13-14 is not logical. Please check your reference and/or your manuscript for typos.

l.238-39: I thought that most of your premise was based upon the presence of n-3 fatty acids in the Calunus oil as you focus on n-3 fatty acid effects throughout your introduction and discussion.  If indeed you believe that the wax esters are significantly involved in determining the impact of the Calunus oil on BMI, then the study should be done with a group of subjects given ONLY  n-3 fatty acids with exercise as a control, to determine whether the addition of the other components of Calunus oil such as the wax esters produces significantly different results.  Your results for BMI in Table 3 do suggest an effect of the Calunus oil on BMI, but is this due to the wax esters, n-3 fatty acids or what?

341: please check the reference cited to be sure it supports your statement in the manuscript

Author Response

Dear reviewer,

thank you for your revision. We put our anwers in italics after/below your points.

l.44: Please check your reference #10 to determine what type of lifting was done by the subjects to calculate 1-RM (e.g.; bench press, squat, or what) ? Also I would like to know the characteristics of the subjects, as 1-RM, in my best judgement, is only accurate and valid when testing advanced weight trainers. If the subjects were not in that category, I would skip the reference to that test or choose a better reference.

Reference #10 fits the scope of our manuscript, bringing an interesting results. Authors Smith et al. used not only 1-RM testing with leg press, chest press, knee extension, and knee flexion on machines but also hand-grip dynamometry and standard isokinetic dynamometry testing of knee extensors and flexors. Moreover, before the testing, they included a period of familiarization.

l.51: Please indicate that the data collected in Ref.’s 13 and 14 was collected using a mouse model.

We corrected this.

l.140 My understanding is that RPE is on a scale of 1-10, so 13-14 is not logical. Please check your reference and/or your manuscript for typos.

We used original 15-grade Borg RPE scale which the subjects were familiar with. Reference added.

l.238-39: I thought that most of your premise was based upon the presence of n-3 fatty acids in the Calunus oil as you focus on n-3 fatty acid effects throughout your introduction and discussion.  If indeed you believe that the wax esters are significantly involved in determining the impact of the Calunus oil on BMI, then the study should be done with a group of subjects given ONLY n-3 fatty acids with exercise as a control, to determine whether the addition of the other components of Calunus oil such as the wax esters produces significantly different results. 

We believe that wax esters play a significant role in the Calanus oil effects. However, this study was designed to prove the effect of Calanus oil, but not particularly wax esters. In this time, we are not able to specify which part of Calanus has an effect, however this was not the aim of this study. We agree with the reviewer that to investigate the effect of wax esters should be warranted in further studies. Indeed, in that case another design of the study should be applied.

Your results for BMI in Table 3 do suggest an effect of the Calunus oil on BMI, but is this due to the wax esters, n-3 fatty acids or what?

Actually, BMI was decreased in placebo group only.

341: please check the reference cited to be sure it supports your statement in the manuscript 

More details were added in the text concerning this reference.

Reviewer 3 Report

Major Comments:

Statistical Analysis: As the authors performed non-parametric statistics, it is inappropriate to report means – instead the authors should consider reporting medians for their data. In addition, the authors should strongly consider inserting effect sizes. Dosing: Can you please specify whether it was 230mg of EPA and 230 mg of DHA per day? Or was it 230mg combined of EPA and DHA? Did you monitor adherence to ensure that participants were in fact consuming their supplement? Did you screen for dietary fish intake? Was this an unblinded, single-blinded, or a double-blinded trial? Please specify Methods: Overall, the methods are poorly written and do not merely provide enough detail. Methods should be written in such a manner that the experiment could be – hypothetically – executed by another research group. I would strongly encourage providing more detail throughout the methods. Please see some edits/suggestions below.

Minor Comments:

Did you have any hypotheses for the current study? Functional fitness tests: The authors need to do better describing the standard operating protocol for their measurements. The methods should be described in such a manner that another individual can be recreated by another individual. When were the outcome measures conducted throughout the trial? The authors need to specify if this was a pre-post design or something else. Bioelectrical Impedance: How was visceral fat measured in this study? Bioelectrical impedance is not able to distinguish visceral fat from subcutaneous fat. Please address as this is not referred to in methods. I would recommend referring this as, “Body Fat”. In addition, Bioelectrical impedance does not provide information on, “skeletal muscle mass” (Table. 3) – instead, it measures lean body mass. Please consider changing.

Line Comments:

12: please remove, “the study aimed to investigate” with “we investigated”

21 – 23: You should specify which outcome measures were improved with the Calanus Oil.

25: Please insert a concluding statement.

29: Please provide reference

30-32: Please rephrase this sentence, confusing to the reader

32-33: Please define the terms “non-functional tissue” and  “compensate” in regard to this statement

35: Please define “complications” in regard to this statement

37: Please be more specific than “general health”, what are the desired health benefits ?

38: Please define “desirable adaptive response” or elaborate

39: Recommend not using the word “except” to begin this sentence

39-41: References are needed to back up this statement

48: Please define what a “substantial” amount is

49: Please state the full term for both EPA and DHA the first time they are mentioned in the

article, after which the acronyms are perfectly acceptable. For example, “…bioavailable source

of eicosapentaenoic acid (EPA) and docosahexaenoic acid (DHA) for…”

53: Please define “effectiveness” or reword this sentence to be more clear

53: Please provide references for the studies being referred to

53: Please be specific as to which “randomized control trial” you are referring to

56: Please be more specific than “an exercise adaptation”

56-57: Please elaborate as to whether reduced inflammation or immune system improvements are the mechanism of focus (if both, please rephrase this sentence, confusing to the reader)

62: Please change “randomized placebo trial” to “randomized control trial”

72-73: If only 53 subjects completed the study and were included in analysis please reflect this in your abstract.

72: Please replace “dropped out” with “withdrew” and state if their reason(s) for doing so were

related to the intervention.

73-75: Please be more clear when these results were collected, is this referring to baseline?

90: Please define “W” before using the abbreviation

93: Please define what the authors mean by “individually chosen slope”

95: Please remove the “r” between “reached” and “more”

94-96: Please be more specific as to when the exercise test was stopped, was it RER > 1.06 and pedal cadence ? or simply the first of the two to be present?

97: Please define RPM before using abbreviation

98: Please be more specific as to what the “appointed level” referred to in this sentence is meaning

100: Please be clear which exercise test these values were recorded, was it the EST or ramp protocol ?

114: Chair stand test is not a measure of “strength” but rather functionality, please consider rephrasing this

139: Please define “TRX”

140: Please provide reference for RPE scaling used

148: 60-85% of VO2peak for these untrained participants is likely above their lactate threshold

and is thus a difficult intensity to maintain for this duration. Please provide reasoning for how

they were able to maintain this pace for 40 minutes.

162: Please confirm the differences referred to here were measured at which timepoint of the study

175-176: Please provide p-value for the decrease since the authors have noted “non-significant decrease in visceral fat”

178: Please be specific as to which “variables” the authors are referring to

184-185: Please be consistent with use of “standard deviation” or “SD” in figure descriptions

187: Please be consistent with the use of “VFA” the authors have interchangeably used the abbreviation to describe both visceral fat and visceral fatty acids.

187: Why is there a 9 at the top left corner of the table?

189-191: Please provide references for the statements made in this sentence

193: Please define “age-related complications”

198: Please provide values for safe and recommended dosage of N-3 PUFA supplementation previously acknowledged

200: Please define “beneficial effects”

204: Please define “omega 3 index”

211: Please define “dynamic and explosive strength” what tests were used to assess this?

214-216: Please provide a reference for this statement

225: Recommend saying the specific test and not “lower extremity strength”

235: Comment mentioned previously, please ensure BIA devices are specific to measuring visceral fat before concluding this statement

242: Please define “specific variables”

243-245: Positive correlation in upper and lower body strength does not determine complexity of training program, please consider rephrasing this sentence

252-254: Please provide reference here

Table 2: Please change “(number)” for both Chair Stand and Arm Curl, to “(repetitions)”.  

Author Response

Dear reviewer, thank you for your valuable comments. We made a lot of changes in the manuscript, esp. in the method and results sections.

Below are answers to you comments in italics.

Major Comments:

Statistical Analysis: As the authors performed non-parametric statistics, it is inappropriate to report means – instead the authors should consider reporting medians for their data. In addition, the authors should strongly consider inserting effect sizes.

We corrected this and we have used median and IQR instead of mean and SD now. We have added the effect size.  

Dosing: Can you please specify whether it was 230mg of EPA and 230 mg of DHA per day? Or was it 230mg combined of EPA and DHA? Did you monitor adherence to ensure that participants were in fact consuming their supplement? Did you screen for dietary fish intake? Was this an unblinded, single-blinded, or a double-blinded trial?

230 mg was combined dose of EPA and DHA, the content was specified for each fatty acid (as 125mg EPA + 105 mg DHA) as is now written in the manuscript.

The adherence was monitored by omega-3 index measurement in red blood cells (RBC), this was slightly, but significantly increased. Dietary fish intake, and generally consumption of omega-3 was monitored by questionaire. This information was added into the manuscript.

The study was randomized double blinded-study. This information was included into manuscript.

Please specify Methods: Overall, the methods are poorly written and do not merely provide enough detail. Methods should be written in such a manner that the experiment could be – hypothetically – executed by another research group. I would strongly encourage providing more detail throughout the methods.

We corrected this.

Please see some edits/suggestions below.

Minor Comments:

Did you have any hypotheses for the current study? Functional fitness tests: The authors need to do better describing the standard operating protocol for their measurements. The methods should be described in such a manner that another individual can be recreated by another individual. When were the outcome measures conducted throughout the trial? The authors need to specify if this was a pre-post design or something else.

We specified the Senior Fitness Test in detail and implemented the design more visibly.

Bioelectrical Impedance: How was visceral fat measured in this study? Bioelectrical impedance is not able to distinguish visceral fat from subcutaneous fat. Please address as this is not referred to in methods. I would recommend referring this as, “Body Fat”. In addition, Bioelectrical impedance does not provide information on, “skeletal muscle mass” (Table. 3) – instead, it measures lean body mass. Please consider changing.

InBody 720 offers SMM as well as VFA as outcomes. We have added comments that those outcomes were probably not as exact as if they were measured by DXA.

Line Comments:

12: please remove, “the study aimed to investigate” with “we investigated”

We did this.

21 – 23: You should specify which outcome measures were improved with the Calanus Oil.

This is specified in the last sentence – „group with Calanus supplementation experienced significantly higher improvement of functional strength of lower body and specification (chair stand test) was added.

25: Please insert a concluding statement. – A concluding statement inserted.

29: Please provide reference – A reference was added. On the other hand, the statement is so well known, that does not require any reference in our opinion.

30-32: Please rephrase this sentence, confusing to the reader  - The sentence was corrected.

32-33: Please define the terms “non-functional tissue” and  “compensate” in regard to this statement – The whole statement was removed.

35: Please define “complications” in regard to this statement – Potentional adverse effects on PA in the elderly were added.

37: Please be more specific than “general health”, what are the desired health benefits ? It was corrected to „muscle mass and function“.

38: Please define “desirable adaptive response” or elaborate. It was corrected to „desirable anabolic response“.

39: Recommend not using the word “except” to begin this sentence. It was corrected.

39-41: References are needed to back up this statement – The references, which support this statement are given here and in the next sentences.

48: Please define what a “substantial” amount is – It was specified to „(up to 80-90% of total lipdis)“.

49: Please state the full term for both EPA and DHA the first time they are mentioned in the article, after which the acronyms are perfectly acceptable. For example, “…bioavailable source of eicosapentaenoic acid (EPA) and docosahexaenoic acid (DHA) for…” It was corrected as suggested.

53: Please define “effectiveness” or reword this sentence to be more clear – It was corrected to „greater efficacy“.

53: Please provide references for the studies being referred to -  The reference is listed in the next sentence and also other reference included.

53: Please be specific as to which “randomized control trial” you are referring to – It was corrected to „study of Da Boit at al.“.

56: Please be more specific than “an exercise adaptation” – It was corrected to „effect on muscle but also on other organs“.

56-57: Please elaborate as to whether reduced inflammation or immune system improvements are the mechanism of focus (if both, please rephrase this sentence, confusing to the reader) – It was rephrased to „An exercise and omega-3 effect on muscle but also on other organs might be driven through the mechanism of reduced inflammation and improvements in the immune system“.

62: Please change “randomized placebo trial” to “randomized control trial”. We corrected this.

72-73: If only 53 subjects completed the study and were included in analysis please reflect this in your abstract. We corrected this.

72: Please replace “dropped out” with “withdrew” and state if their reason(s) for doing so were. related to the intervention. We corrected this.

73-75: Please be more clear when these results were collected, is this referring to baseline? Yes, results are based on the pre-post test (baseline = before the program and measurements after the program). We corrected this in the manuscript.

90: Please define “W” before using the abbreviation. We corrected this.

93: Please define what the authors mean by “individually chosen slope”. We apologize for a missing value in a range. Now it should make sence. The slope was between 15 and 30 Watts per minute and this was individually chosen by the sport physician (based on his experience and according to expected fitness level of each subject).

95: Please remove the “r” between “reached” and “more”. We corrected this.

94-96: Please be more specific as to when the exercise test was stopped, was it RER > 1.06 and pedal cadence ? or simply the first of the two to be present? The test was until exhaustion, i.e. inability to keep the cadence and according to RER.

97: Please define RPM before using abbreviation. We corrected this.

98: Please be more specific as to what the “appointed level” referred to in this sentence is meaning. We corrected this to make it more clear.

100: Please be clear which exercise test these values were recorded, was it the EST or ramp protocol ? The exercise stress test involved one grade of submaximal exercise and ramping protocol to peak effort. The peak values were used for analysis. We believe that the text describes this.

114: Chair stand test is not a measure of “strength” but rather functionality, please consider rephrasing this. We tried to rephrase.

139: Please define “TRX”. We corrected this.

140: Please provide reference for RPE scaling used. We used original 15-grade Borg RPE scale which the subjects were familiar with. Reference added.

148: 60-85% of VO2peak for these untrained participants is likely above their lactate threshold and is thus a difficult intensity to maintain for this duration. Please provide reasoning for how they were able to maintain this pace for 40 minutes. For most of the time, the walking speed was corresponding rather to lower level of above-mentioned range. During the entire walking trips, there were several short sections of higher intensity (i.e. interval training) according to actual situation (weather, beginning of intervention program vs end of intervention program, etc.). It was added to the manuscript.

162: Please confirm the differences referred to here were measured at which timepoint of the study. These values are from baseline of the study which was written int the Table heading. We made it more clear in the text as well.

175-176: Please provide p-value for the decrease since the authors have noted “non-significant decrease in visceral fat”. That has been done.

178: Please be specific as to which “variables” the authors are referring to. We corrected this.

184-185: Please be consistent with use of “standard deviation” or “SD” in figure descriptions. We corrected to medians and IQR.

187: Please be consistent with the use of “VFA” the authors have interchangeably used the abbreviation to describe both visceral fat and visceral fatty acids. There was mistake in the table which we corrected. VFA means visceral fat area.

187: Why is there a 9 at the top left corner of the table? This is mistake, we apologize for that.

189-191: Please provide references for the statements made in this sentence. This sentence serves as an opening of disccusion section based on general knowledge which is widely accepted.

193: Please define “age-related complications”. We believe this is clear from the introduction.

198: Please provide values for safe and recommended dosage of N-3 PUFA supplementation previously acknowledged. – Information on safety dosage of omega-3 was added.

200: Please define “beneficial effects”. We rephrased this.

204: Please define “omega 3 index”. The formulation “The omega-3 index is defined as the sum of EPA and DHA in erythrocyte membranes expressed as a percentage of total fatty acids“ was added into the manuscript.

211: Please define “dynamic and explosive strength” what tests were used to assess this? Specific test were added.

214-216: Please provide a reference for this statement. This does not need a reference as this is a result of our study.

225: Recommend saying the specific test and not “lower extremity strength”. We corrected this to „functional lower body strength“.

235: Comment mentioned previously, please ensure BIA devices are specific to measuring visceral fat before concluding this statement. As it was mentioned above.

242: Please define “specific variables”. We changed to some variables.

243-245: Positive correlation in upper and lower body strength does not determine complexity of training program, please consider rephrasing this sentence. It was correlation between these improvements (i.e. changes post-pre) not the strength values.

252-254: Please provide reference here. The reference was added.

Table 2: Please change “(number)” for both Chair Stand and Arm Curl, to “(repetitions)”.  We corrected this.